# A super-resolution scanning algorithm for lensless microfluidic imaging using the dual-line array image sensor

**Dian Tian**[ID]**, Ningmei Yu**[ID]**\*, Zhengpeng Li, Shuaijun Li, Na Li**

School of Automation and Information Engineering, Xi'an University of Technology, Xi'an, Shaanxi Province, China

\* yunm@xaut.edu.cn

**Data Availability Statement:** All relevant data are within the paper.

**Funding:** This work was supported by the National Natural Science Foundation of China (No. 61771388). The funder had no role in study design,

## Abstract

The lensless optical fluid microscopy is of great significance to the miniaturization, portability and low cost development of cell detection instruments. However, the resolution of the cell image collected directly is low, because the physical pixel size of the image sensor is the same order of magnitude as the cell size. To solve this problem, this paper proposes a super-resolution scanning algorithm using a dual-line array sensor and a microfluidic chip. For dual-line array sensor images, the multi-group velocity and acceleration of cells flowing through the line array sensor are calculated. Then the reconstruction model of the super-resolution image is constructed with variable acceleration. By changing the angle between the line array image sensor and the direction of cell flow, the super-resolution image scanning and reconstruction are achieved in both horizontal and vertical directions. In addition, it is necessary to study the row by row extraction algorithm for cell foreground image. In this paper, the dual-line array sensor is implemented by adjusting the acquisition window of the image sensor with a pixel size of 2.2μm. When the tilt angle is 21 degrees, the equivalent pixel size is 0.79μm, improved 2.8 times, and after de-diffraction its average size error was 3.249%. As the angle decreases, the image resolution is higher, but the amount of information is less. This super-resolution scanning algorithm can be integrated on the chip and used with a microfluidic chip to realize on-chip instrument.

## Introduction

Collecting and analyzing cell images of biological tissues is an important basis for disease diagnosis, health monitoring, and new drug development in medicine today [1,2]. Flow cytometry can quickly and accurately perform cell detection. However, its promotion and application are harmed by cost and portability. With the popularization of concepts such as smart medicine and telemedicine, the lensless optical fluid microscope technology for miniaturization, automation, and low cost of cell image acquisition instruments were proposed in 2006 [3]. Since the pixel size of the image sensor and cell size is on the same order of magnitude as cell size, the resolution of the image collected by the lensless optical fluid microscope is low. Then the method of passing the target through a special aperture array is proposed to reduce the pixel size and achieve super-resolution imaging to solve this problem [4,5].

data collection and analysis, decision to publish, or preparation of the manuscript.

**Competing interests:** The authors have declared that no competing interests exist.

Scholars from all over the world are trying to solve the problem of low resolution of the imaging results of lensless systems by implementing super-resolution reconstruction. A method of real super-resolution reconstruction by generating a micro-lens effect above or on the surface of the object has been proposed [6,7]. In order to obtain more details of cells, the multi-angle micro-displacement of the optical path is generated, and the cells are scanned for micro-displacement [8,9]. Then the high-resolution image is synthesized into a group of low-resolution sub-pixel-shifted images. But at the same time, an accurate optical path system is required, and the implementation cost is high. Similarly, the fluid flow first generates low-resolution images of multiple frames of targets and then reconstructs a single super-resolution image through a multi-frame super-resolution algorithm [10–12]. Different from this, the convolutional neural network structure is improved to establish the feature mapping relationship between low-resolution images and high-resolution images [13–15]. The multi-wavelength phase recovery and multi-angle light source diffraction tomography was used to realize the high-resolution imaging of the lensless system and restores the depth image [16,17]. Also, an up-sampling phase retrieval scheme is proposed to bypass the resolution limit of the pixel size of the imager [18]. This method introduces some optical devices and improves the resolution through the corresponding phase recovery algorithm. Our research team has proposed a method of super-scan imaging using a single-line array detector, which sets an oblique linear array image sensor under the microfluidic channel to scan the flowing cells. After reconstruction, a super-resolution scan of the cells can be obtained. Compared with the area array image sensor, its method greatly reduces the power consumption occupied by the pixel unit. However, this method requires very high control accuracy of the cell flow rate, and the reconstructed image is easily distorted.

In this article, our proposed solution is to build a super-resolution scanning system using a dual-line image sensor. It can accurately calculate cell flow velocity and acceleration. Firstly, two single-line array detectors with micro-pitch and parallel structure are adopted to construct the double-line array structure. The time difference between the cells flowing through two independent linear array sensors is used to accurately calculate the instantaneous flow velocity and acceleration of the cells. Secondly, the single-line scan imaging process is re-modeled, and the transformation relationship between the line scan image and the object image coordinates is pushed to reconstruct the line scan image and restore the super-resolution image of the object. In addition, the foreground separation of the line scan image, speed calculation, and other issues have been studied in depth. Based on the mean background modeling, a multi-threshold foreground coarse segmentation method is proposed to update the background model, and the foreground model of the line scan image is extracted by the background model. Feature detection and feature matching algorithms are used to match the time difference and displacement difference of cells as they pass through two linear array sensors, and accurately calculate the instantaneous flow velocity and acceleration information of the cells.

## Materials and methods

### System structure and basic model

The system structure (Fig 1A) of the dual-line array image sensor consists of a 405nm laser plane wave source, a microfluidic chip, and a CMOS plane array image sensor MT9P031. In the system, when the pixel size of the image sensor is smaller, the resolution of the reconstructed image will be higher and the image will be clearer. However, the current commercial linear array image sensor has too large pixels, so we choose a smaller pixel and a high sampling rate area array image sensor. This sensor can adjust the size of image acquisition window through the function of a region of interest (ROI), so as to replace the two-wire array sensor, and its pixel size is 2.2μm. In this function, only the pixel reading of the window area will be activated, so the row rate can be greatly

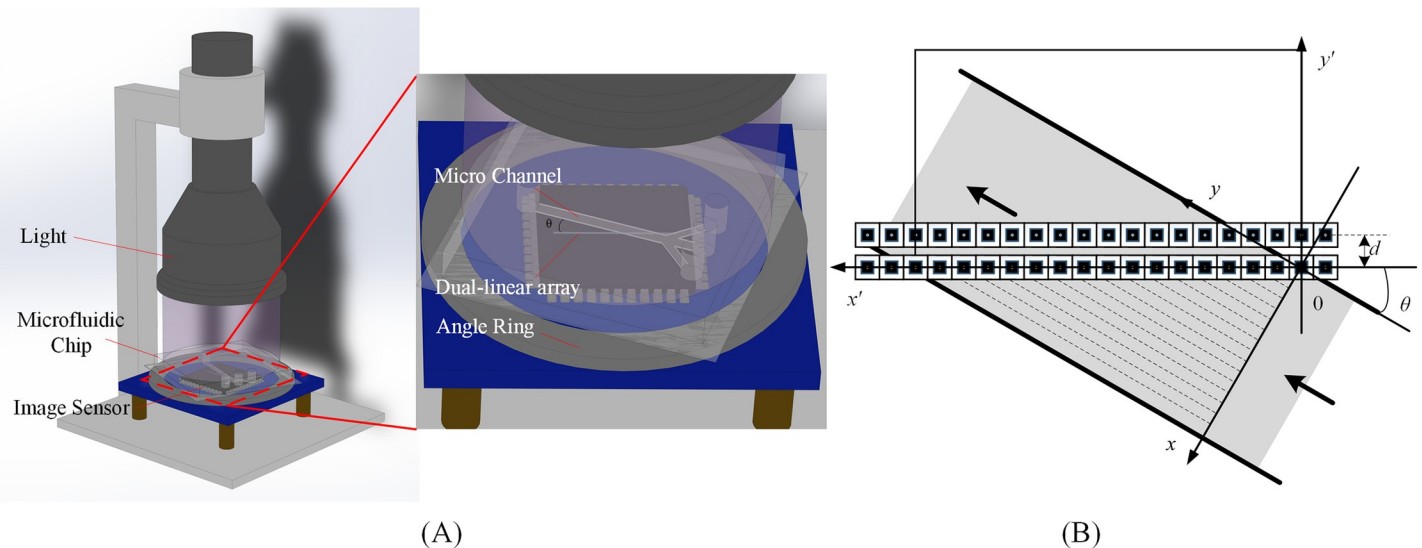

**Fig 1. The structure and imaging principle.** (A) The system structure of the dual-line array image sensor. (B) The process of acquisition by the dual-line array sensor.

improved. The schematic diagram of the dual-line array sensor structure is shown in Fig 1B, and the two linear array sensors are placed in parallel with a space of $d$. When the linear array sensor is at an acute angle to the direction of cell flow, the lateral resolution of scanning imaging can be increased, and that is the principle of super-resolution scanning imaging. In this section, the basic mathematical modeling of its structure will be carried out, including the establishment of the coordinate system, speed model, resolution model and distance model.

As shown in Fig 1B, the acquisition resolution in the inclined placement mode is smaller than that in the vertical placement mode. Taking the first intersection of the object flow direction and the linear array sensor as the origin, the object flow direction of the channel as the axis $y$, the flow direction as the positive direction, the direction perpendicular to the channel as the axis $x$, and the direction pointing to the channel one measurement as the positive direction, the rectangular coordinate system of the channel object image is established, named $C_1$. By a similar process, the rectangular coordinate system of the linear scanning image, called $C_2$, is established. Special attention should be paid to the fact that the intersection of the axis $x'$ and the axis $y'$ is not the zero point of the axis $y'$, but the coordinate of the axis $y'$ when the cell is passing through the linear array sensor.

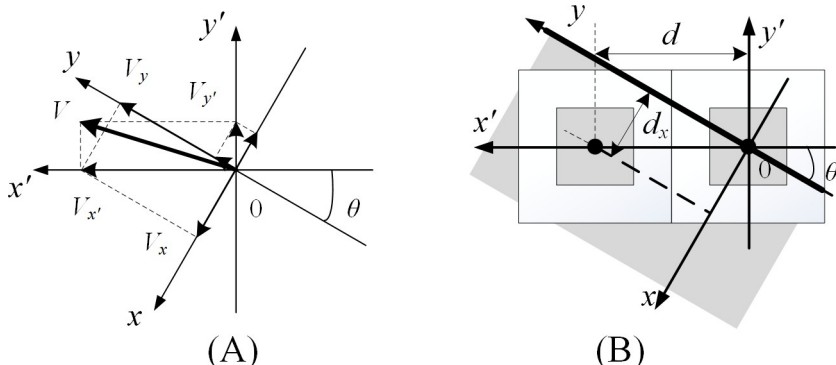

**Fig 2. Some model of the tilted linear array sensor.** (A) The velocity decomposition model. (B) The super-resolution model.

Suppose that there is an object flowing at a speed $V$ in the channel, as shown in Fig 2A. As calculated using Eq (1), the velocity $V_x$, $V_y$, $V_{x'}$ and $V_{y'}$ can be obtained by decomposing the velocity in the coordinate system $C_1$ and $C_2$ respectively. Similarly, the transformation relationship of acceleration between two coordinate systems is determined in Eq (2).

$$\begin{cases} V_{x'} = V_y \cdot \cos(\theta) + V_x \cdot \sin(\theta) \\ V_{y'} = V_y \cdot \sin(\theta) - V_x \cdot \cos(\theta) \\ V_x = V_{x'} \cdot \sin(\theta) - V_{y'} \cdot \cos(\theta) \\ V_y = V_{x'} \cdot \cos(\theta) + V_{y'} \cdot \sin(\theta) \\ \theta = \arctan(V_x / V_y) \end{cases} \tag{1}$$

$$\begin{cases} a_{x'} = a_y \cdot \cos(\theta) + a_x \cdot \sin(\theta) \\ a_{y'} = a_y \cdot \sin(\theta) - a_x \cdot \cos(\theta) \\ a_x = a_{x'} \cdot \sin(\theta) - a_{y'} \cdot \cos(\theta) \\ a_y = a_{x'} \cdot \cos(\theta) + a_{y'} \cdot \sin(\theta) \end{cases} \tag{2}$$

Fig 2B is an enlarged view of the intersection of axes. When the pixel spacing of the linear array sensor is $d$, the imaging resolution of the axis $x'$ direction is $d$, that of the axis $x$ direction is $d_x$. Then the transformation formula between $d$ and $d_x$ can be deduced in Eq (3). To ensure that the scale of the reconstructed image is the same as that of the real object, the resolution in the axis $x$ direction should be equal to the resolution in the axis $y$ direction.

$$\begin{cases} d_x = d \cdot \sin(\theta) \\ d_y = d_x \end{cases} \tag{3}$$

When the imaging sample reaches the origin, the linear array sensor starts acquiring images, and suppose there is a point $P_1$ on the object this time. As shown in Fig 3, its coordinate is $(x,y)$. The distance from the axis $x$ and the axis $y$ is $S_x$ and $S_y$. If the object without a lateral velocity flows, the pixel corresponding to point $P_1$ on the linear array sensor is $L_1$, else is $L_2$ and the lateral flow distance is $S_{V_x}$. Then the coordinate of the corresponding point $P_2$ on the linear scanning image is $(x',y')$. The ordinate $y'$ represents the number of frames since the object starts imaging when $P_1$ is acquired. The true distance between point $P_2$ and the axis $y$ is $S_{x'}$. According to the relationship between imaging resolution, pixel size and pixel coordinates, the Eq (4) can be obtained.

$$\begin{cases} S_x = x \cdot d_x \\ S_y = y \cdot d_y \\ S_{x'} = x' \cdot d \end{cases} \tag{4}$$

## Methods of cell foreground extraction

When the linear array sensor scanning is used to image cells in microfluidics, the influence of background impurities in microfluidics can be avoided, and only the dynamic change information of cells flowing can be collected. However, because of the noise of image sensor pixels and the non-uniformity of the light source, the uneven fringe noise will be formed on the

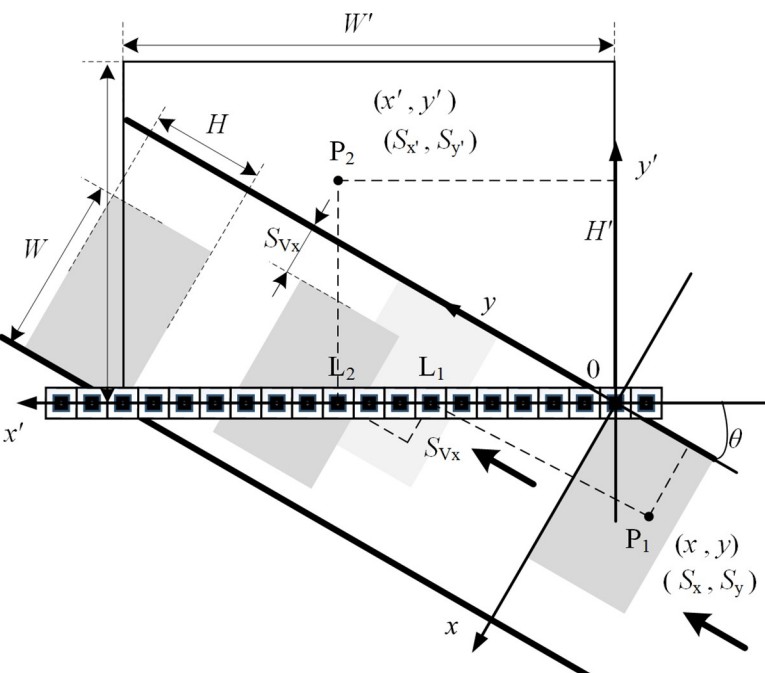

**Fig 3. The distance calculation model in the case of the tilted linear array sensor.**

scanned image. After the system is started, the noise will remain stable. In the linear scanning image, the pixels and light intensity of each line are the same. As a result, for a continuous, short period of time, it can be considered that the background is almost the same. Based on this assumption, the pixel value of background can be obtained by simple mean modeling that build without cells flow. Further in this paper, the background model is updated in real-time by identifying pixels with cells flow through the multi-threshold method, which reduces the interference to the background model.

Firstly, $i$ is the current number of collected times, and when the sensor first collects, $i$ is 1. N rows of background images, from $i$-$N$ to $i$-1, are buffered to establish a background initial mean model. $\mathbf{P}_i$ is the line pixel value of the $F_i$ row collected currently, and $\bar{\mathbf{P}}_i$ is the mean value of the rows from $F_{i-N}$ to $F_{i-1}$. Meanwhile, $P_{i,j}$ is a pixel value of the column j in the $F_i$ row, so the formula of background initial mean model is

$$\bar{P}_{i,j} = \frac{1}{N} \sum_{k=i-N}^{i-1} P_{k,j} \tag{5}$$

Then the initial foreground difference information $\mathbf{EP}_i$ of line $F_i$ will be obtained after line $F_i$ is cached in Eq (6).

$$\mathbf{EP}_i = \mathbf{P}_i - \bar{\mathbf{P}}_i \tag{6}$$

Based on this information, the background mask $\mathbf{MP}_i$ of line $F_i$ is

$$\mathbf{MP}_i = \begin{cases} 0 & \mathbf{EP}_i < T_1, \mathbf{EP}_i > T_2 \\ 1 & \text{other} \end{cases} \tag{7}$$

where $T_1$ and $T_2$ are the lower and the upper threshold of the background, respectively. The pixels in which cells are present will be filtered by this mask, and the new value of background

mean model $\bar{\mathbf{P}}_i'$ of rows $F_{i-N+1}$ to $F_i$ will be re-calculated by

$$\bar{\mathbf{P}}_i' = \frac{\bar{\mathbf{P}}_i \cdot (N-1) + \mathbf{MP}_i \cdot \mathbf{P}_i}{N-1+\mathbf{MP}_i} \tag{8}$$

Finally, the new foreground difference information $\mathbf{EP}_i'$ of cells is

$$\mathbf{EP}_i' = \mathbf{P}_i - \frac{\bar{\mathbf{P}}_i \cdot (N-1) + \mathbf{MP}_i \cdot \mathbf{P}_i}{N-1+\mathbf{MP}_i} \tag{9}$$

## Methods of instantaneous velocity

The accuracy of cell velocity calculation determines the distortion of reconstructed scanned images. In lensless imaging, when the object is far from the imaging surface, the light will be diffracted through the object to form a diffraction ring, each ring of gray level is uniform. Therefore, the maximally stable extremal regions (MSER) algorithm is used to detect the alternating light and dark diffraction rings, and then the feature points on the boundary of the maximally stable extremal regions are screened out. Finally, the scanning matching of feature points in another linear array sensor acquisition image is carried out by the sum of squared differences (SSD) algorithm. Then the set of feature points on two linear array sensors is obtained for calculating cell flow velocity.

MSER, similar to the watershed, can detect connected regions such as diffraction rings in cell diffraction images. Under the obvious detection effect, we compressed the dynamic range of the image before MSER detection to reduce the calculation time. According to the characteristics of diffraction rings, the corner features are mostly distributed in the upper and lower vertex positions of the MSER region. So the coordinates of the Corner point are quickly determined, its ordinate is the extremum of the MSER area's ordinate, and the value of its abscissa is the mean value of the abscissa at the extremum of the ordinate. Then each MSER region can extract two corner features, and select the appropriate corners to match. It is necessary to analyze the coordinates of each corner after the initial extraction of corner features. Each corner point must be not too close, and a corner point with a more obvious corner feature should be selected while the distance is relatively close. The minimum distance between the corners' ordinates can be determined by extracting the maximum difference of the ordinates of corners. The corner points with more obvious features can be screened by the calculation matrix of the corner point features in Eq (10).

$$V_{corner} = \text{abs}(\text{sum}(M_{data} .* M_{corner})) \tag{10}$$

$M_{data}$ is a window matrix of the corner point, and $M_{corner}$ is a corner feature calculation matrix, which is related to the actual line array direction and is obtained by experiments. $M_{corner}$ and $M_{data}$ are the same size, and the larger the $V_{corner}$, the more obvious the feature.

Assume that $K$ feature points are extracted from the scanned image of the first linear array sensor, and an image of size $(H+1)\times(W+1)$ is extracted around the feature point $k$. The feature point, that is the center point of this image, is denoted as $M_{L1}(0,0,k)$. Then the SSD matching algorithm on the image of the second linear array sensor is

$$V_{SSD}(i,j,k) = \sum_{m=-H/2}^{H/2} \sum_{n=-W/2}^{W/2} \left( M_{L2}(i+m, j+n) - M_{L1}(m,n,k) \right)^2 \tag{11}$$

where $M_{L2}(i,j)$ is the pixel value of the coordinates $(i,j)$ on the scanned image of the second linear array sensor, and $V_{SSD}(i,j,k)$ is the SSD value of the pixel point and the feature point $k$ on the scanned image of the first linear array sensor. The large the SSD value, the higher the

matching degree between the two feature points. During the search process, the feature point with the largest SSD value is selected as the final matching point.

The difference in the displacement of cell images acquired by the dual-line array sensors is small because of the short distance between the first and the second linear array sensor. So an SSD matching search area of the second linear array sensor is set up based on the coordinates of the feature points of the first linear array sensor, to reduce the search efficiency of the SSD matching algorithm. Assume that the line rate of the line array sensor is $f$, the pixel size is $s_{pixel}$, and the line array pitch is $d$. The coordinates of two adjacent feature points on the first linear array sensor are $(x_i, y_i)$, $(x_{i+1}, y_{i+1})$, and the coordinates of corresponding matching points on the second linear array sensor are $(x'_i, y'_i)$, $(x_{i+1}', y_{i+1}')$. Then the time difference between the first and the second linear array sensor of the point on the cell is $(y'_i - y_i)/f$, and the lateral displacement is $x_i - x'_i$. Therefore, the lateral velocity $V_{x'_i}$ and longitudinal velocity $V_{y'_i}$ of this point in the coordinate system $C_2$ are

$$\begin{cases} V_{x'_i} = s_{pixel} \cdot f \cdot (x_i - x'_i)/(y'_i - y_i) \\ V_{y'_i} = d \cdot f/(y'_i - y_i) \end{cases} \tag{12}$$

Similarly, the lateral velocity $V_{x_{i+1}'}$ and longitudinal velocity $V_{y_{i+1}'}$ of this point in the coordinate system $C_2$ can be calculated. Then the lateral acceleration $a_{x'}$ and longitudinal acceleration $a_{y'}$ during this period can be calculated by the two adjacent feature points $V_{x'}$, $V_{y'}$. The time difference between the two feature points on the cell after passing the first linear array sensor is $(y_{i+1} - y_i)/f$, and the acceleration $a_{x'_i}$ of the cell during this period is

$$\begin{cases} a_{x'_i} = f \cdot (V_{x_{i+1}'} - V_{x'_i})/(y_{i+1} - y_i) \\ a_{y'_i} = f \cdot (V_{y_{i+1}'} - V_{y'_i})/(y_{i+1} - y_i) \end{cases} \tag{13}$$

In this way, the velocity and acceleration information of the coordinate system $C_1$ can be obtained. By a similar process, the $K$ velocities and the $K-1$ accelerations of all feature points are calculated.

## The reconstruction with variable acceleration

Suppose that an object flows in the microchannel, with $V_x$ and $V_y$ as the initial velocity of the axis $x$ and axis $y$, $a_x$ and $a_y$ as the acceleration of the axis $x$, and axis $y$. According to the physical relationship of distance, speed and acceleration, the Eq (14) can be obtained

$$\begin{cases} S_{x'} \cdot \sin(\theta) = S_x + S_{V_x} \\ y' \cdot \dfrac{1}{f} \cdot \left( V_y + \dfrac{1}{2f} \cdot y' \cdot a_y \right) = S_y + \dfrac{S_x}{\tan(\theta)} + \dfrac{S_{V_x}}{\tan(\theta)} \end{cases} \tag{14}$$

where

$$S_{V_x} = y' \cdot \frac{1}{f} \cdot \left( V_x + \frac{1}{2f} \cdot y' \cdot a_x \right)$$

Then the coordinate transformation formula of the object coordinate system mapping in the line scanned coordinate system is

$$
\begin{cases}
x' = x + \dfrac{y' \cdot (V_x + a_x \cdot \frac{y'}{2 \cdot f})}{d \cdot f \cdot \sin(\theta)} \\
Ay'^2 + By' + C = 0
\end{cases}
\tag{15}
$$

where

$$
\begin{cases}
A = \dfrac{a_{y'}}{2 \cdot f} \\
B = V_{y'} \\
C = -d \cdot f \cdot \sin(\theta) \cdot (y \cdot \sin(\theta) + x \cdot \cos(\theta))
\end{cases}
$$

So the solution $y'$ of the one-variable quadratic equation can be written as

$$
y' = \frac{-B \pm \sqrt{B^2 - 4AC}}{2A} \quad \text{when } B^2 - 4AC \geq 0
\tag{16}
$$

In reality, it is difficult for small objects to maintain a constant acceleration flow, which is mostly variable acceleration flow. It is assumed that the object is running at speed $V_0$ as shown in Fig 4A, and the linear array sensor starts to acquire at time $t_0$. So the instantaneous flow velocity of the object at $t_1$, $t_2$ and $t_3$ are $V_1$, $V_2$ and $V_3$, respectively. That is mean, the object has three acceleration $a_0$, $a_1$ and $a_3$ for three time periods when flowing through the linear array sensor. In this case, this paper adopts an iterative mapping method to map the acceleration change time in the linear scan coordinate system on the object coordinate system, so that the different acceleration areas of the object correspond to the linear scan area one by one. Then the object coordinate system maps the reconstructed image on the line scan coordinate system. As shown in Fig 4B, it is a schematic diagram of an object passing through a linear array sensor at this speed change, which respectively shows the position of the linear array sensor on the object at each time point. At the moment, the object and the linear array sensor intersect at two points a and b respectively, and the area between the original point on the object and the two points a and b flows through the linear array sensor with $V_0$ as the initial speed and $a_0$ as the acceleration. Similarly, the area between the four points a, b, c, and d on the object flows through the line array sensor with $V_1$ as the initial speed and $a_1$ as the acceleration. If the object

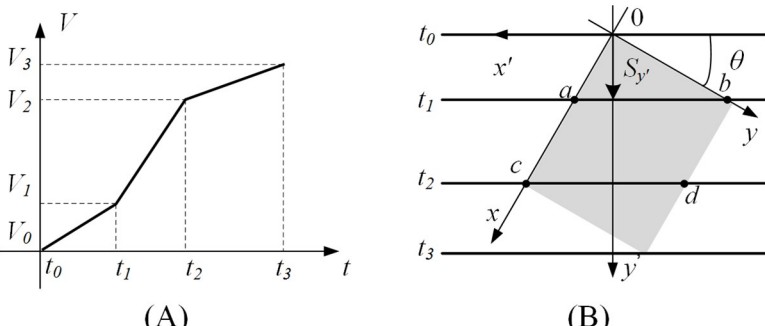

**Fig 4. The situation with the variable speed.** (A) Object flows at the variable speed. (B) The position of the line array sensor on the object.

collects data from line $y_0$ from $t_0$ to $t_1$, the distance the object moves along the axis $y'$ is

$$S_{y'_0} = V_{y'_0} \cdot y'_0 \cdot \frac{1}{f} + \frac{1}{2} \cdot a_0 \cdot \left( y'_0 \cdot \frac{1}{f} \right)^2 \tag{17}$$

Then the distance from the point $b$ to the axis $x$ is

$$b_{y_0} = \frac{S_{y_0}}{d_y} = \frac{S_{y'_0}}{d \cdot \sin(\theta) \cdot \sin(\theta)} = \frac{S_{y'_0}}{d \cdot \sin^2(\theta)} \tag{18}$$

Considering the linear array sensor as a straight line, the equation of the straight line at time $t_1$ can be obtained by the slope of the linear array sensor in the coordinate system $C_1$.

$$y_0 = -\frac{1}{\tan(\theta)} \cdot x_0 + b_{y_0} = -\frac{1}{\tan(\theta)} \cdot x_0 + \frac{S_{y'_0}}{d \cdot \sin^2(\theta)} \tag{19}$$

Similarly, the equation at time $t_2$ is

$$y_0 = -\frac{1}{\tan(\theta)} \cdot x_0 + b_{y_0} = -\frac{1}{\tan(\theta)} \cdot x_0 + \frac{S_{y'_1}}{d \cdot \sin^2(\theta)} \tag{20}$$

where

$$S_{y'_1} = S_{y'_0} + V_{y'_1} \cdot (y'_1 - y'_0) \cdot \frac{1}{f} + \frac{1}{2} \cdot a_1 \cdot \left( (y'_1 - y'_0) \cdot \frac{1}{f} \right)^2$$

Then the three acceleration regions can be mapped to the coordinate system of the object.

$$\begin{cases} a = a_0 & x/\tan(\theta) + y \le b_{y_0} \\ a = a_1 & b_{y_0} < x/\tan(\theta) + y \le b_{y_1} \\ a = a_2 & b_{y_1} < x/\tan(\theta) + y \end{cases} \tag{21}$$

To generalize it to $K$ accelerations, the Eq (22) can be written as

$$b_{y_i} = \frac{S_{y_i}}{d_y} = \frac{S_{y'_i}}{d \cdot \sin^2(\theta)}, i = 1, 2, 3, \ldots, K \tag{22}$$

where

$$S_{y'_i} = S_{y'_{i-1}} + V_{y'_i} \cdot (y'_i - y'_{i-1}) \cdot \frac{1}{f} + \frac{1}{2} \cdot a_i \cdot \left( (y'_i - y'_{i-1}) \cdot \frac{1}{f} \right)^2$$

The Eq (23) can be written as

$$a = a_i \text{ when } b_{y_i} < x/\tan(\theta) + y \le b_{y_{i+1}}, i = 1, 2, 3, \ldots, K \tag{23}$$

Then the coordinates of the object coordinate system are mapped to the linear scanning imaging coordinate system, and the speed information of the corresponding area is brought into the corresponding pixel value. By solving the quadratic equation of each acceleration region, the coordinates $(x'_i, y'_i)$ of the linear scan coordinate system, mapped by the coordinates $(x_i, y_i)$ of the object coordinate system, can be obtained. It should be noted that $(x'_i, y'_i)$ are the coordinates relative to each acceleration segment, and the coordinates $(x''_i, y''_i)$ in the coordinate

system $C_2$ should be

$$\begin{cases} x_i^{''} = x_i^{'} \\ y_i^{''} = \sum_{s=0}^{i} y_i^{'} \end{cases} \quad (24)$$

Then the pixel value of coordinates $(x_i, y_i)$ can be calculated from the pixels around the coordinate $(x_i^{''}, y_i^{''})$.

## Results and discussion

### Analysis of cell foreground extraction

We used 20μm microspheres as test objects, and the angle of the dual-line array sensor is 21 degrees. When the number of acquisition lines of the sensor is set to 10 lines, the frame rate is 1230fps. The flow rate of the solution is related to the sampling rate of the image sensor. When the sampling rate of the image sensor is higher, the more samples can be processed per unit time. After considering these issues, this paper chooses a suitable flow rate of solution, which is 5 μL/min ~ 10 μL/min. We extract an image every 10 frames in Fig 5, and the microsphere flowed in 0.12s. In our system, only two lines of pixels are used to reconstruct super-resolution images. This chapter explains the foreground extraction of the scanned image.

Fig 6A is the image of 500 lines which are scanned pixels from the first linear array sensor. Due to the unevenness of the light source, there are vertical stripes with uneven brightness and width on the scanned image. When the number of cache lines N is taken as 20, the background mean model and the algorithm in this paper are tested. Fig 6B is the extracted microsphere scanned image by the background mean model, and the background part does not eliminate the uneven noise very well, although the microsphere foreground part can be separated. Differently, the pixel value of foreground difference information is firstly assigned 1, when is between 15 and -15, otherwise is 0, as shown in Fig 6C. Obviously, it roughly divides the white background part and the black foreground part. After avoiding the influence of the foreground part on the background mean model, the microsphere scanned image is shown in Fig 6D. Compared with the image extracted by the background mean model, the algorithm proposed in this paper has less background noise. Therefore, background interference can be largely reduced, and cell images can be more accurately extracted.

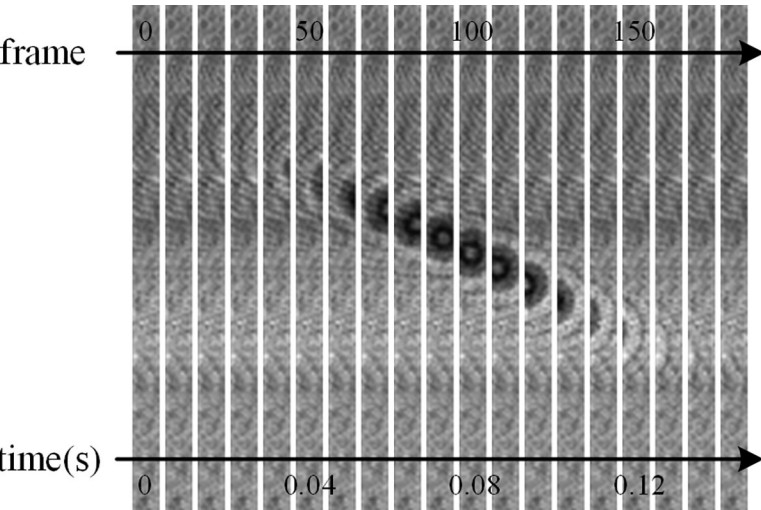

**Fig 5. Test results of cell foreground extraction for 20μm microspheres.**

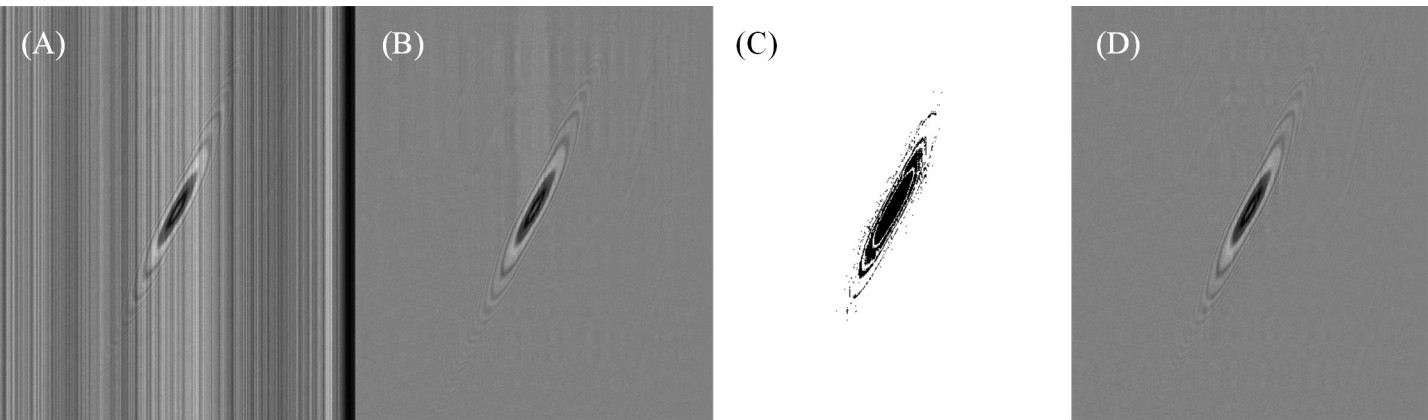

**Fig 6. Test results of cell foreground extraction for 20μm microspheres.** (A) The 500 lines of raw images from the first linear sensor are connected to one image. (B) The extracted microsphere image directly by background mean model. (C) The mask image with the threshold of plus-minus 15. (D) The foreground image extracted by the improved method.

## Analysis of speed calculation

The detection effect of MSER is mainly affected by the parameters of $A_{max}$ and $\Delta$. Of course, the limitations of the connected regions also can be further filtered. In this paper, the $A_{max}$ is set to a larger value of 20 to detect more diffraction rings. First, the dynamic range of the image is compressed in 10 steps, when $A_{max}$ is 20 and $\Delta$ is 1. As shown in Fig 7A, each color is an MSER area. And only the first and second diffraction rings can be detected, when the dynamic range is below 190. When above 190, the third-order diffraction ring can appear. However, after 200, there will be too many subdivided detection areas, which will increase the processing burden. Therefore, the dynamic range of the diffraction image collected by scanning imaging can be compressed to 190, and then better MSER detection can be performed. Second, $\Delta$ is tested in steps of 0.3 in Fig 7B, when the dynamic range is below 190. With the increase of $\Delta$, the detection effect of diffraction ring becomes worse. And with the decrease of $\Delta$, more and more detection areas are subdivided. When between 2 and 2.3, the detection effect is in an intermediate state.

Finally, corner features are extracted from MSER features directly in the first left of Fig 8 when the dynamic range is 190 and $\Delta$ is 2. According to the scanning image of the experiment structure and through the experimental analysis, the window radius of $M_{corner}$ is 4 pixels in Eq (25) and the results are shown in Fig 8 on the far right. As you see, our method can extract and screen better corner features that meet the requirements easily and quickly.t

$$M_{corner} = \begin{bmatrix} 0 & 0 & 0 & 0 & 0 & 1 & 1 & 1 & 0 \\ 0 & 0 & 0 & 0 & 0 & 1 & 1 & 1 & 0 \\ 0 & 0 & 0 & 0 & 0 & 1 & 1 & 0 & 0 \\ 0 & 0 & 0 & 0 & 0 & 0 & 0 & 0 & 0 \\ 0 & 0 & 0 & 0 & 0 & 0 & 0 & 0 & 0 \\ 0 & 0 & 0 & 0 & 0 & 0 & 0 & 0 & 0 \\ 0 & 0 & -1 & -1 & 0 & 0 & 0 & 0 & 0 \\ 0 & -1 & -1 & -1 & 0 & 0 & 0 & 0 & 0 \\ 0 & -1 & -1 & -1 & 0 & 0 & 0 & 0 & 0 \end{bmatrix} \tag{25}$$

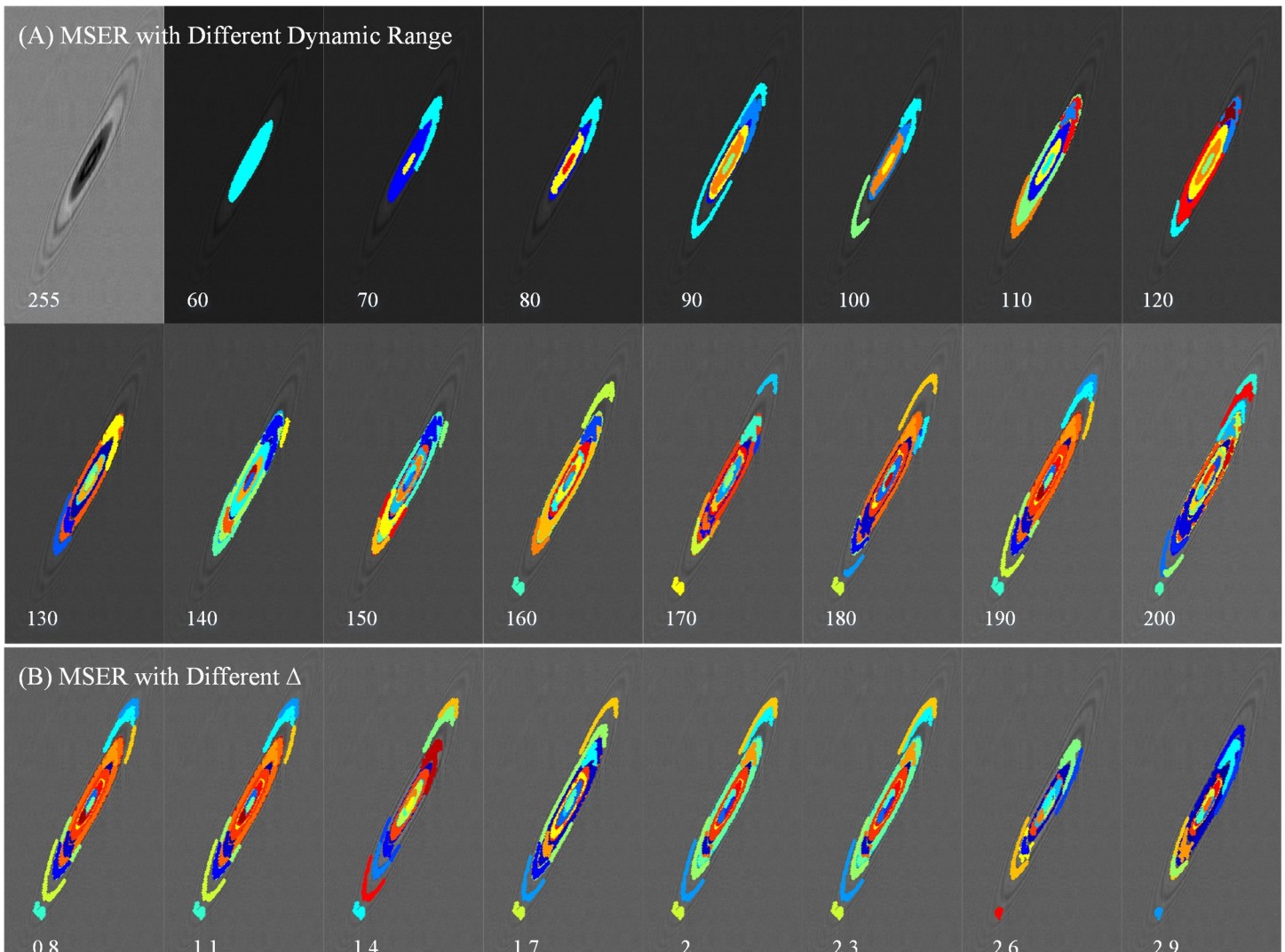

**Fig 7. The analysis of MSER.** (A) MSER with the different values of image dynamic range. (B) MSER with the different Δ.

To reduce the search efficiency of the SSD matching algorithm, this paper sets up the SSD matching search area of the second linear array sensor based on the feature point coordinates of the first linear array sensor. Using this algorithm when $H = W = 20$, 11 feature points are detected and matched on the scanning image (Fig 9A). The first and third lines are the images corresponding to the feature points on the first linear array, and the second and fourth lines are the images corresponding to the feature points on the second linear array. As you see the feature points matching is accurate in Fig 9B.

Then the velocities of all 11 feature points and 10 acceleration information, shown in Table 1, can be calculated by applying formulas (1), (2), (12), (13). When the microspheres flow in the channel, the maximum change of the transverse velocity along the channel direction is about 216 μm/s, and the longitudinal velocity is about 514 μm/s. It is obvious that the horizontal and vertical velocity are not stable in the process of cell flow, and the instantaneous velocity of cell flow can be accurately calculated using the method presented in this paper.

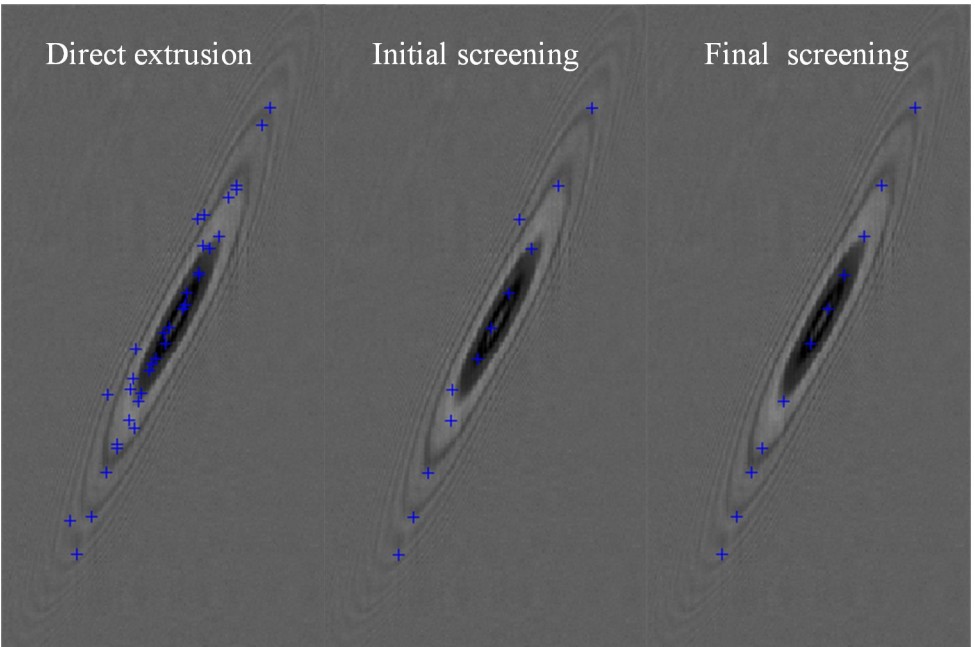

**Fig 8. The feature point after extracted and screened from MSER when the value of the image dynamic range is 190, $A_{max}$ is 20 and $\Delta$ is 2.**

## Reconstruction results

Having gathered the information of the velocity of the microsphere, the scanned image of the 20μm microsphere was reconstructed. To compare the superiority of the algorithm of the variable acceleration, we use the algorithm of the variable acceleration and the uniform velocity to reconstruct the microsphere, and the results are shown in Fig 10A and 10B. Significantly, the latter is much better than the former, and the multi-order diffraction ring of the 20μm

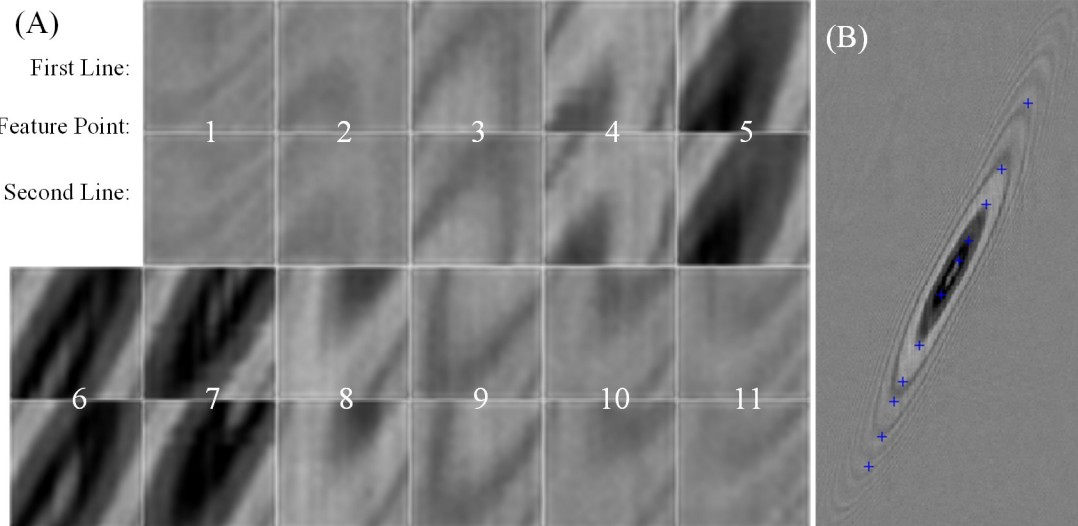

**Fig 9. The analysis of the matching feature points.** (A) The feature maps of this point in the image of the dual-line array sensor by the SSD algorithm. (B) The corresponding feature points in the image of the second linear array sensor.

**Table 1. Velocity and acceleration information of each characteristic point of the microsphere.**

| Feature point | $V_{x'}(\mu m/s)$ | $V_{y'}(\mu m/s)$ | $a_x(\mu m/s^2)$ | $a_{y'}(\mu m/s^2)$ | $V_x(\mu m/s)$ | $V_y(\mu m/s)$ |
|---|---|---|---|---|---|---|
| 1 | 1040.771 | 416.308 | 15.679 | -831.264 | -7.852 | 1120.917 |
| 2 | 996.431 | 386.571 | -128.123 | 6792.612 | -7.292 | 1040.852 |
| 3 | 1353.003 | 541.200 | 4695.305 | -5043.000 | -10.208 | 1457.192 |
| 4 | 1353.002 | 451.000 | -14346.766 | 15409.167 | 73.773 | 1424.280 |
| 5 | 1353.003 | 676.500 | 13324.547 | -11346.750 | -136.180 | 1506.561 |
| 6 | 1476.003 | 492.000 | -475.913 | -2909.423 | 80.480 | 1553.760 |
| 7 | 1248.925 | 416.308 | -3624.000 | 1641.213 | 68.098 | 1314.720 |
| 8 | 1127.502 | 451.000 | 0.000 | 0.000 | -8.507 | 1214.327 |
| 9 | 1127.502 | 451.000 | -38.049 | 2017.200 | -8.507 | 1214.327 |
| 10 | 1230.003 | 492.000 | 5728.734 | 2881.714 | -9.280 | 1324.720 |

microsphere with little distortion, can be observed from the reconstruction image. This shows that the algorithm of the variable acceleration is necessary in the case that the actual flow direction of the cell is variable and the speed is variable. The resolution is related to the pixel size and the tilt angle of the linear array sensor. According to the Eq (3), the equivalent pixel size is 0.79μm, which is 2.78 times higher than the pixel size of the area array sensor. When the microsphere is collected by the area array sensor with the same pixel size 2.2μm, its resolution is very low as shown in in Fig 10C. The details are much fuzzier than that in Fig 10B, when we enlarge the image to 2.78 times in Fig 10D.

The reconstructed super-resolution image is a diffraction image of the microsphere, and the image of the microsphere can be recovered by the de-diffraction algorithm. This algorithm

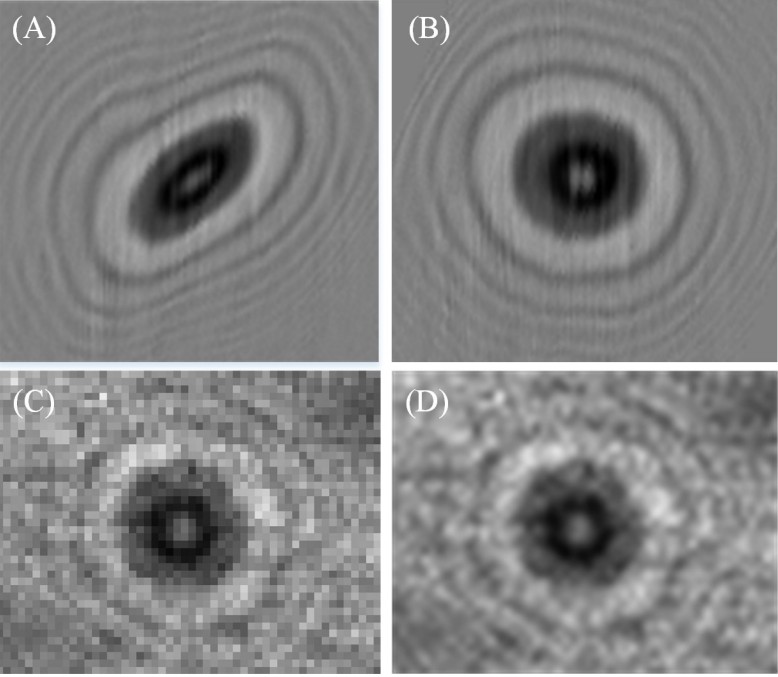

**Fig 10. The comparison of the reconstructed image of the dual-line array sensor and the image of the area array sensor.** (A) The image reconstructed by the algorithm derived from the uniform velocity hypothesis. (B) The image reconstructed by the algorithm derived from the variable acceleration hypothesis. (C) The area array sensor image. (D) 2.78x magnification of the image (C) using a cubic spline interpolation algorithm.

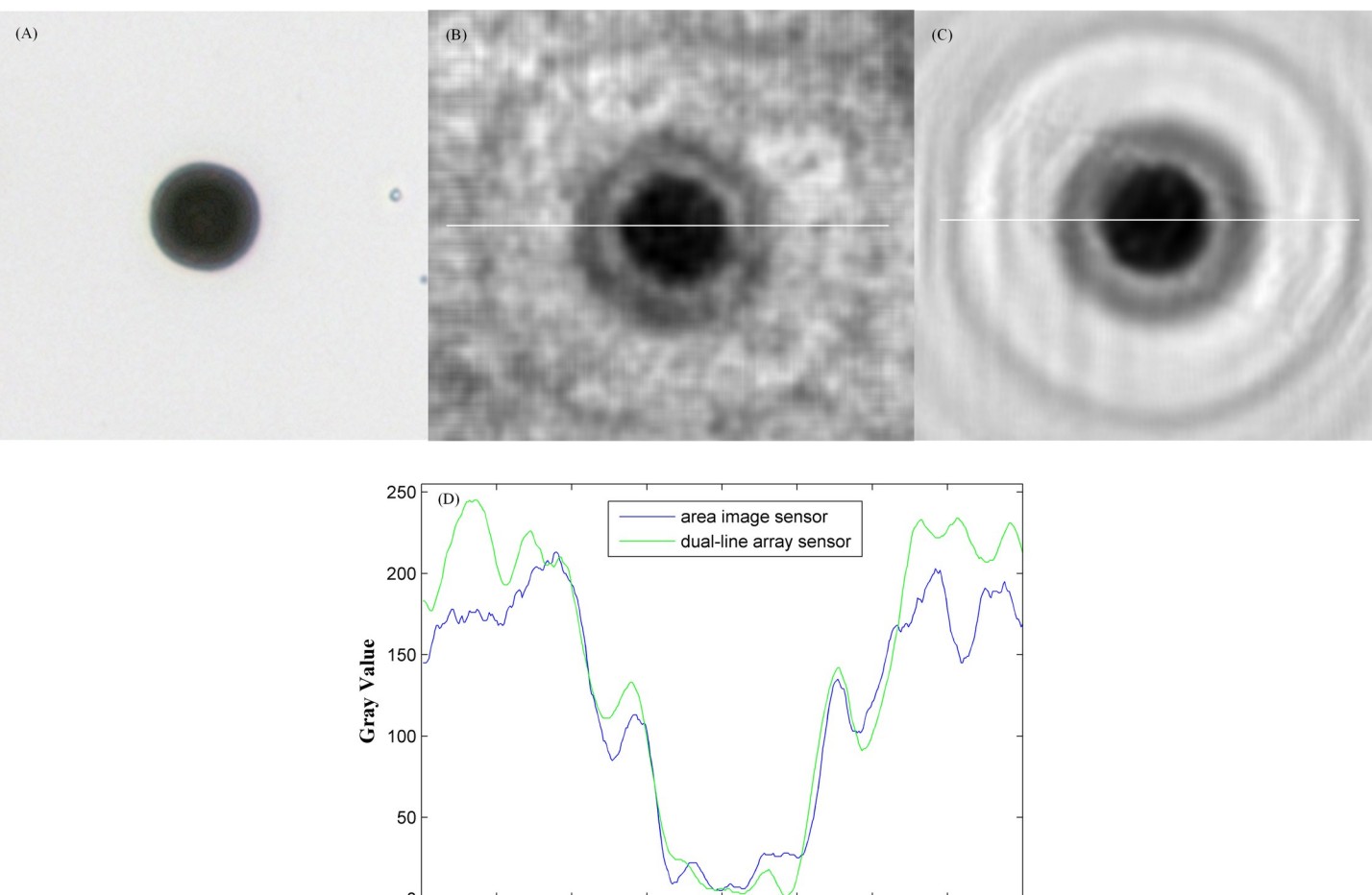

**Fig 11. The analysis of de-diffraction image.** (A) The microscope image under a 10x microscope. (B) The de-diffraction image of Fig 10D. (C) The de-diffraction image of Fig 10B. (D) The pixel values of the white line in (B) and (C) are plotted.

has been studied in the paper [19], and is cited directly here. Fig 11A shows an image of a 20μm microsphere under a 10x microscope. After being magnified four times, the de-diffracted image of Fig 10D is shown in Fig 11B, and the de-diffracted image of Fig 10D is shown in Fig 11C. The pixel values of the white line are plotted in Fig 11D, by comparing these images, the recovery results in this paper has smoother edges and clearer details.

We calculated the peak signal-to-noise ratio (PSNR) and the structural similarity index (SSIM) about the enlarged image of area array sensor and the de-diffraction image of dual-line array sensor. In Table 2, because the ideal image is used as the reference image, the PSNR and SSIM of all images are relatively low. However, the de-diffraction image of the dual-line array

**Table 2. The PSNR and contrast of the image of the different sensor, before and after de-diffraction.**

| Image | PSNR (dB) | SNR (dB) | SSIM |
|---|---|---|---|
| Bilinear Interpolation | 11.945 | 10.623 | 0.053 |
| Cubic Spline Interpolation | 11.981 | 10.659 | 0.054 |
| Dual-line Array Sensor | 19.301 | 17.979 | 0.210 |

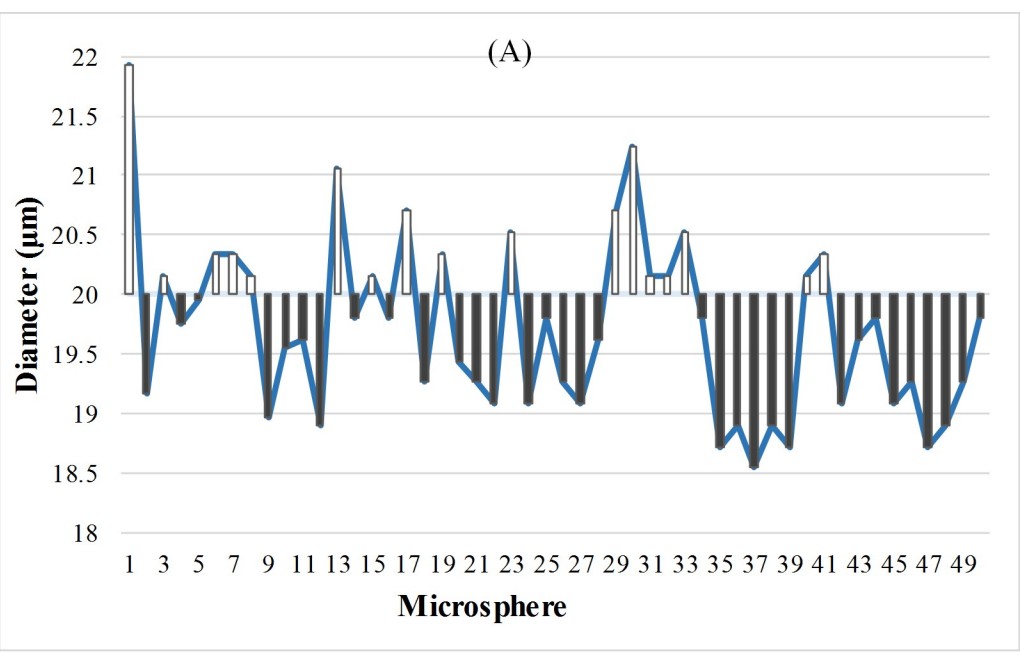

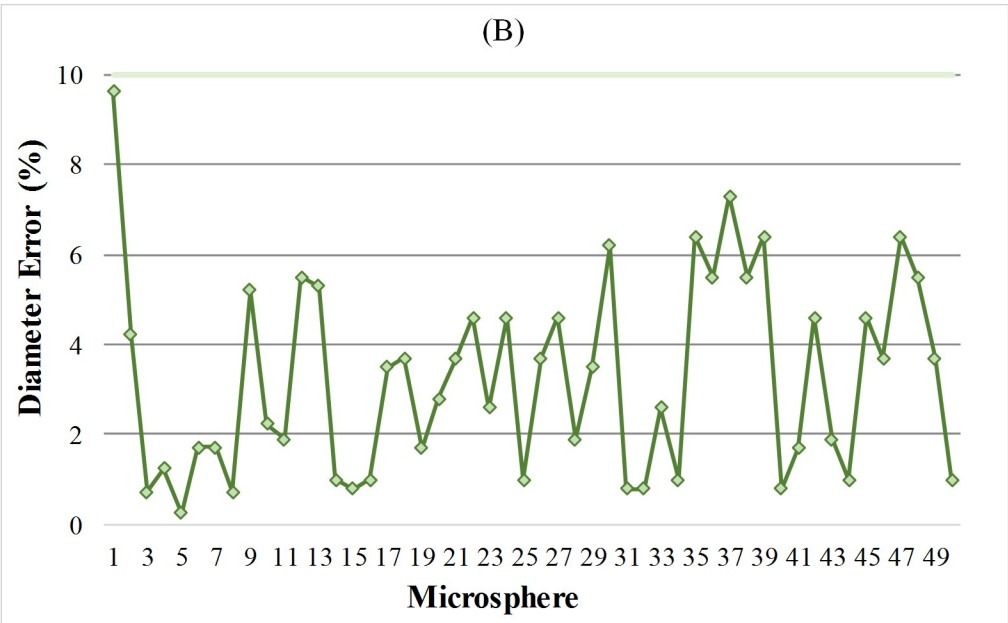

**Fig 12. The analysis of the 50 microsphere images of dual-line array sensor.** (A) The dimensions. (B) The dimensions error.

sensor has a higher PSNR (Improved 1.62 times), and its SSIM is closest to 1 (Improved 3.96 times).

After de-diffraction, the size of this microsphere is 20.7375μm, and the error of microsphere size in our experiment is less than 10%. We calculated the size and its error of 50 microsphere images of dual-line array sensor in Fig 12. Their diameter calculated is shown in Fig 12A, and the white column represents the more part, the black column represents the less part. So it can be seen that the error between the calculated diameter and the real diameter is small, almost within 2μm. Meanwhile, the diameter error of each microsphere is also calculated in Fig 12B,

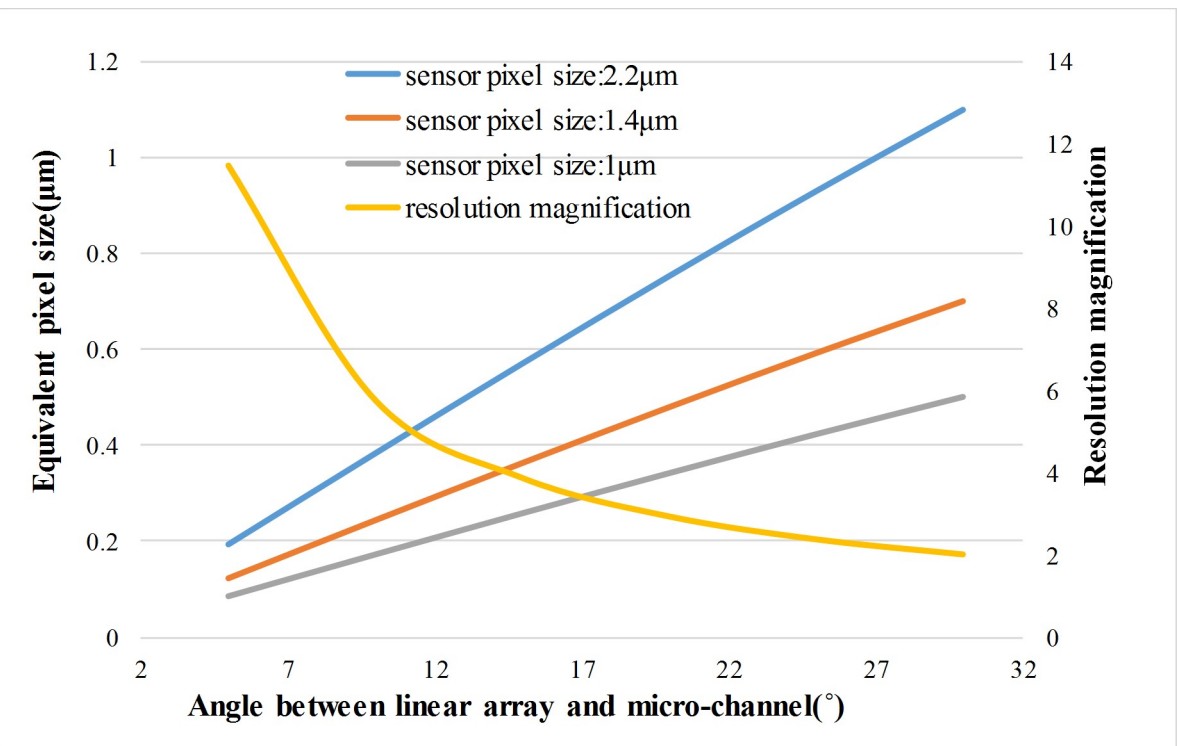

**Fig 13. The relationship between the equivalent pixel size and the angle between the linear array and the micro-channel, as the different pixel size of the dual-line array sensor.**

and the reconstructed dimensions error are within the error range of the microspheres, and the average error is 3.249%.

In the above test, the angle between the micro-channel and the linear array sensor is 21 degrees, and the sensor pixel size is 2.2μm. If a smaller pixel sensor or a smaller angle is used, the method in this paper is still applicable, and the equivalent pixel size is smaller, as shown in Fig 13. The resolution magnification is only related to the tilt angle, not the pixel size.

We have done the same experiment when the angle is 15 or 10 degrees and the pixel size is 2.2μm (Fig 14). As the angle is 15 degrees, the equivalent pixel size is 0.569μm. As the angle is 10 degrees, the equivalent pixel size is 0.382μm. It is smaller than the equivalent pixel size of 0.775μm in paper [8] and 0.770μm in paper [20] by the pixel size image sensor (1.67μm).

The quantity of information can be evaluated by image entropy, and the higher the value, the more information. In Table 3, with the decrease of angle, the image entropy becomes less and less at the real size. When these image is enlarged to same size, the image entropy decreases greatly. This means that when the angle is smaller, the image resolution is higher, but the amount of information is less. Therefore, the tilt angle can be selected to meet you different needs conveniently, just rotating the microfluidic chip.

## Conclusion

In summary, the super-resolution scanning system, using the dual-line array image sensor, is demonstrated to obtain the super-resolution image of cells. Firstly, the method, combined by background mean model and a multi-threshold foreground coarse segmentation method, is designed to extract the cells foreground information from the line of scanning image. Secondly, the multiple sets of velocities and accelerations of cells passing through linear array

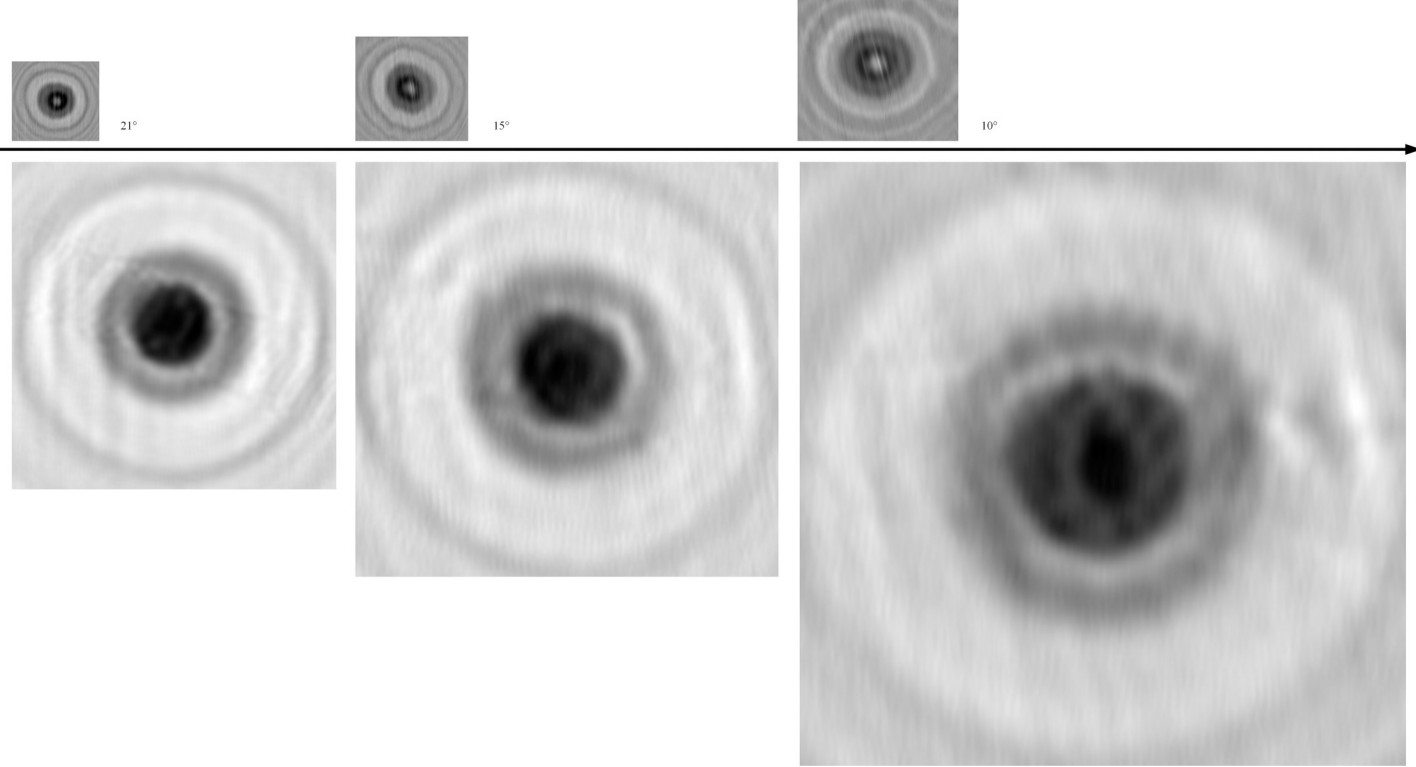

**Fig 14. The super-resolution image reconstructed and its de-diffraction image as the different angles between the linear array and the micro-channel.**

sensors can be calculated with the MSER and SSD algorithm. Then the reconstruction model of scanning image is deduced with uniform speed, uniform acceleration and variable acceleration flow. Finally, the super-resolution image of the cells can be reconstructed. When the pixel size of the linear array sensor is 2.2μm and the angle is 21 degrees, the equivalent pixel size is 0.79μm (Improved 2.8 times, and improved 2.15 times in paper [8,20]). After de-diffraction, the size error of 20μm microsphere was 3.249%, and the PSNR was improved 1.62 times, the SSIM was improved 3.96 times. With the same system structure, the equivalent pixel size can be 0.382μm as the angle is 10 degrees, but the image entropy also decreases. Furthermore, the resolution and the flow rate of solution can be improved by using image sensors with smaller pixels and higher sampling rates, or using the high-throughput microfluidic chips of the multi-channel, and high-throughput analysis can be achieved in the paper [21]. Therefore, it is sufficient to demonstrate that the proposed super-resolution scanning algorithm and system is

**Table 3. The image entropy in the different angles.**

| Image size | Angle | Image entropy (bit/pixel) |
|---|---|---|
| Real size | 10˚ | 1.6469 |
| | 15˚ | 2.1064 |
| | 21˚ | 2.3829 |
| Be enlarged to same size | 10˚ | 1.6469 |
| | 15˚ | 1.6455 |
| | 21˚ | 1.6176 |

effective. The application of the algorithm in lensless optical fluid microscopy can provide a more convenient method of cell detection instruments.

## Author Contributions

**Methodology:** Dian Tian, Ningmei Yu.

**Project administration:** Dian Tian, Ningmei Yu.

**Software:** Dian Tian.

**Supervision:** Ningmei Yu.

**Validation:** Dian Tian, Zhengpeng Li, Shuaijun Li, Na Li.

**Visualization:** Dian Tian.

**Writing – original draft:** Dian Tian, Zhengpeng Li.

**Writing – review & editing:** Dian Tian, Ningmei Yu, Zhengpeng Li, Shuaijun Li, Na Li.

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
