## [Decision Letter · Decision Letter 0]

29 May 2020

PONE-D-20-10408

A super-resolution scanning algorithm for lensless microfluidic imaging using the dual-line array image sensor

PLOS ONE

Dear Dr. Yu,

Thank you for submitting your manuscript to PLOS ONE. After careful consideration, we feel that it has merit but does not fully meet PLOS ONE’s publication criteria as it currently stands. Therefore, we invite you to submit a revised version of the manuscript that addresses the points raised during the review process, in particular points 1-4 raised by reviewer 2.

We look forward to receiving your revised manuscript.

Kind regards,

Christof Markus Aegerter

Academic Editor

PLOS ONE

Journal Requirements:

2. We note you have included a table to which you do not refer in the text of your manuscript. Please ensure that you refer to Table 3 in your text; if accepted, production will need this reference to link the reader to the Table.

Additional Editor Comments (if provided):

Reviewers' comments:

Reviewer's Responses to Questions

**Comments to the Author**

1. Is the manuscript technically sound, and do the data support the conclusions?

Reviewer #1: Yes

Reviewer #2: Yes

2. Has the statistical analysis been performed appropriately and rigorously? 

Reviewer #1: N/A

Reviewer #2: Yes

3. Have the authors made all data underlying the findings in their manuscript fully available?

Reviewer #1: Yes

Reviewer #2: Yes

4. Is the manuscript presented in an intelligible fashion and written in standard English?

Reviewer #1: Yes

Reviewer #2: Yes

5. Review Comments to the Author

Reviewer #1: The manuscript studies an optofluidic on-chip holographic microscope using an angular tilted dual linear sensors. Using this framework the authors demonstrate super-resolution of the on-chip holography, which is one of the limitations of lensfree imaging. The tradeoff in the current implementation is the overall throughput of the device (for example, in comparison to: https://www.nature.com/articles/s41377-018-0067-0). I think that the authors should add a discussion on the throughput / super-resolution for the suggested system.

Also, although the overall English level is OK, I suggest to have this manuscript be proofed by someone that is not involved in the research, to have it written in a more concise manner.

Reviewer #2: The manuscript reports a super-resolution scanning system with a tilted dual-line image sensor. It can reconstruct the cell images that bypass the pixel size limit. The dual line array are used to accurately track the cell velocity when flowing through the channel. This manuscript is in general techanically sound. I recommend its publication if the authors can better address following concerns:

1) The sensor, in fact, is based on the MT9P031 area sensor chip. The authors only read out two lines of the chip. It is unclear to me whether this two line scheme has a real benefit compared to the orginal subpixel optofluidic microscope developed at Yang’s group at Caltech, where hundreads of lines are read out also at a high speed.

2) From line 120 to 124, the author tries to declare a formula for the initial background calculation. However, these parameters are not well defined. I assume Fi is the row number which should belong to N rows, i∈N. If so, the meaning of Fi-N and Fi-1 are unclear to me. What is the range of index i? The parameter k in the equation (5) hasn’t been defined either.

3) What is the distance bewteen the channel and the sensor chip? In the original optofluic microscope device, the cover glass was removed and the channel is directly placed on top of the sensing area. If the distance is large, did they propagate the light to the sample plane using a phase retrieval process?

4) For the reconstructed results in Fig 10. As the author demonstrates, the result of variable acceleration (Fig 10 B) is much better than the uniform speed (Fig 10 A). I am curious about this performance difference. Does that mean the current system could only perform well in a certain condition? I hope the author could discuss it.

5) Only 20-um microsphere is demonstrated in the manuscript. This reviewer feel that the impact may be a little short for the lab-on-a-chip community. It will be better if the authors can images of certain cells.

6) I also suggest the authors to give a better introduction and review of the super-resolution lensless microscopy approach that bypasses the pixel size limit in their second paragraph. New development in this field include the use of up-sampling phase retrieval process to bypass the pixel size limit, see, for example, “Wide-field, high-resolution lensless on-chip microscopy via near-field blind ptychographic modulation,” Lab on a Chip, 20, 1058 - 1065, 2020.

6. PLOS authors have the option to publish the peer review history of their article (what does this mean?). If published, this will include your full peer review and any attached files.

Reviewer #1: No

Reviewer #2: No

---

## [Author Response · Author response to Decision Letter 0]

4 Jun 2020

Additional Editor Comments Response:

1. The bold words mean the sentences have been revised in the “Revised Manuscript with Track Changes” file and the “Manuscript” file.

2. Table 3 does not refer in the text of my first manuscript. I'm sorry that this is an error that should not have occurred. It has been modified correctly in the revision file.

3. My figure files has been uploaded to the Preflight Analysis and Conversion Engine (PACE) digital diagnostic tool, and ensured that figures meet PLOS requirements.

4. We have updated the data availability statement.

Notice: The “Revised Manuscript with Track Changes” file is different from the “Manuscript” file in the deleted sentences and the added sentences, the former reserved these sentences with red and the latter did not.

Reviewer #1: The manuscript studies an optofluidic on-chip holographic microscope using an angular tilted dual linear sensors. Using this framework the authors demonstrate super-resolution of the on-chip holography, which is one of the limitations of lensfree imaging. The tradeoff in the current implementation is the overall throughput of the device (for example, in comparison to: https://www.nature.com/articles/s41377-018-0067-0). I think that the authors should add a discussion on the throughput / super-resolution for the suggested system.

Also, although the overall English level is OK, I suggest to have this manuscript be proofed by someone that is not involved in the research, to have it written in a more concise manner.

Response: Thanks to the reviewers for their suggestions, we have noticed the mentioned papers and discussed them in the new manuscript. The experiment in the first manuscript shows that the flow rate of solution is about 5μL/min ~ 10μL/min, which is only the data under the single channel of 100μm width. The method used in our paper is to collect as much sub-pixel information as possible through the image sensor's high sampling rate and the flow rate of solution control. The sampling rate of the image sensor and the actual flow rate of the cell affect the setting of the flow rate of solution. In our experiment, we chose a flux that is suitable for the sampling rate of the image sensor. Therefore, our focus is different from the papers presented by the reviewers. We focus on high-resolution imaging of flowing cells, which are smaller in size than the mentioned paper. Furthermore, our system can also increase the flow rate of solution by using image sensors with faster sampling rates or using multi-channel high-throughput microfluidic chips. Finally, the manuscript has been revised by someone good in English.

Reviewer #2:

1) The sensor, in fact, is based on the MT9P031 area sensor chip. The authors only read out two lines of the chip. It is unclear to me whether this two line scheme has a real benefit compared to the original subpixel optofluidic microscope developed at Yang’s group at Caltech, where hundreds of lines are read out also at a high speed.

Response: The subpixel optofluidic microscope developed at the Yang’s group at Caltech, is based on an area array image sensor. The idea is to collect multi-frame low-resolution images of the area array, and then reconstruct high-resolution images by multi-frame super-resolution algorithm.

Our system is based on the dual-line array image sensor. It collects the sub-pixel information of the cell through the dual-line array image sensor, one of which is used as a basic super-scan image, and the other is used to calculate the instantaneous flow rate of the cell at multiple points.

In contrast, the advantages of our system are: 1. The system only samples the flowing cells through two rows of pixels, which can naturally reduce the noise caused by the low cleanliness of the microfluidic chip. 2. The current commercial linear array image sensor has too large pixels, so we choose a smaller pixel and high sampling rate area array image sensor with a region of interest (ROI). Then our research team is also designing the corresponding dual line array image sensor, which means that there are only two rows of pixels and the sampling rate is higher. Reading only two rows of pixels can reduce the area and power consumption caused by too many pixels of the sensor while acquiring higher resolution images. It is more conducive to the development of the lab-on-chip systems and portable mobile monitoring equipment.

2) From line 120 to 124, the author tries to declare a formula for the initial background calculation. However, these parameters are not well defined. I assume Fi is the row number which should belong to N rows, i∈N. If so, the meaning of Fi-N and Fi-1 are unclear to me. What is the range of index i? The parameter k in the equation (5) hasn’t been defined either.

Response: I'm sorry that the description of our manuscript is unclear. In the manuscript, i is the current number of collected times, and when the sensor first collects, i is 1. N rows of background images, from i-N to i-1, are buffered to establish a background initial mean model. For example, assuming that N is 20, when the 1000th acquisition is performed, that is, i=1000, then Fi-N to Fi-1 represent the 980th to 999th rows.

I'm sorry that there is an error in equation (5), where k is the loop variable in the cumulative calculation, and its range is i-N to i-1.

Thanks to the reviewers for your careful inspection, these two issues have been revised in the new manuscript.

3) What is the distance between the channel and the sensor chip? In the original optofluic microscope device, the cover glass was removed and the channel is directly placed on top of the sensing area. If the distance is large, did they propagate the light to the sample plane using a phase retrieval process?

Response: The distance between the channel and the sensor chip, that is, the object plane and the imaging plane is about 600μm. The original optofluidic microscope device mentioned by the reviewers is to remove the cover glass on the image sensor surface to reduce the distance between the object surface and the imaging surface, which can reduce the diffraction. In this paper, during the test, the cover glass was not removed, but the acquired diffraction image was reconstructed, and then the phase information of the object was recovered using the phase retrieval process. Since this algorithm is not the focus of this paper, and our research team has published the corresponding paper, this method is directly cited in Ref.19. It is worth noting that the method in this paper is also applicable to the small distance between the object surface and the imaging surface.

4) For the reconstructed results in Fig 10. As the author demonstrates, the result of variable acceleration (Fig 10 B) is much better than the uniform speed (Fig 10 A). I am curious about this performance difference. Does that mean the current system could only perform well in a certain condition? I hope the author could discuss it.

Response: I'm sorry that the description of our manuscript is unclear. Fig 10A in the text is an image reconstructed by the algorithm derived when the microspheres are assumed to have a uniform velocity. Fig 10B is an image reconstructed by the algorithm derived when the microsphere is assumed to have variable acceleration. The flow velocity of the microspheres is changing and the direction of the flow is also changing. The results of Fig 10B are better than those of Fig 10A, which shows that the algorithm derived when using uniform velocity alone does not fit the real situation well. The algorithm deduced when the microspheres are assumed to be variable acceleration can more accurately reconstruct the image under real conditions. Therefore, this does not mean that the system can only perform well under certain conditions. Instead, this means that the system can compensate for the distortion of the reconstructed image due to the uneven cell flow velocity by estimating the acceleration at different points.

5) Only 20-um microsphere is demonstrated in the manuscript. This reviewer feel that the impact may be a little short for the lab-on-a-chip community. It will be better if the authors can images of certain cells.

Response: This suggestion is very well. We analyzed the 20-um microsphere in the experiment and got the conclusion as described in the paper. In later experiments, we plan to test red blood cells, white blood cells, and algae, and design an embedded, on-chip detection device. However, due to the impact of the COVID-19 epidemic, we cannot return to the laboratory for related experiments. Therefore, we hope to publish the conclusions obtained first.

6) I also suggest the authors to give a better introduction and review of the super-resolution lensless microscopy approach that bypasses the pixel size limit in their second paragraph. New development in this field include the use of up-sampling phase retrieval process to bypass the pixel size limit, see, for example, “Wide-field, high-resolution lensless on-chip microscopy via near-field blind ptychographic modulation,” Lab on a Chip, 20, 1058 - 1065, 2020.

Response: Thanks to the reviewers for providing us with a new paper, we have added an analysis of the paper to the new manuscript. The mentioned paper uses an up-sampling phase retrieval process to bypass the pixel size limit. The mentioned paper introduces some optical devices on the optical level and improves the resolution through the phase recovery algorithm. Our system is based on the original system without increasing the complexity of the system, through the high sampling rate linear array sensor and the flow of the cell itself, to collect more sub-pixel information of the cell to improve the imaging resolution. The research perspectives of the two are different.

---

## [Decision Letter · Decision Letter 1]

10 Jun 2020

A super-resolution scanning algorithm for lensless microfluidic imaging using the dual-line array image sensor

PONE-D-20-10408R1

Dear Dr. Yu,

We’re pleased to inform you that your manuscript has been judged scientifically suitable for publication and will be formally accepted for publication once it meets all outstanding technical requirements.

Kind regards,

Christof Markus Aegerter

Section Editor

PLOS ONE

Additional Editor Comments (optional):

Reviewers' comments:

Reviewer's Responses to Questions

**Comments to the Author**

1. If the authors have adequately addressed your comments raised in a previous round of review and you feel that this manuscript is now acceptable for publication, you may indicate that here to bypass the “Comments to the Author” section, enter your conflict of interest statement in the “Confidential to Editor” section, and submit your "Accept" recommendation.

Reviewer #2: All comments have been addressed

2. Is the manuscript technically sound, and do the data support the conclusions?

Reviewer #2: Yes

3. Has the statistical analysis been performed appropriately and rigorously? 

Reviewer #2: Yes

4. Have the authors made all data underlying the findings in their manuscript fully available?

Reviewer #2: Yes

5. Is the manuscript presented in an intelligible fashion and written in standard English?

Reviewer #2: Yes

6. Review Comments to the Author

Reviewer #2: The authors have addressed the my previous comments. I think the presented results are technically sound and it is fine to accept.

7. PLOS authors have the option to publish the peer review history of their article (what does this mean?). If published, this will include your full peer review and any attached files.

Reviewer #2: No

---

## [Editor Report · Acceptance letter]

16 Jun 2020

PONE-D-20-10408R1 

A super-resolution scanning algorithm for lensless microfluidic imaging using the dual-line array image sensor 

Dear Dr. Yu:

I'm pleased to inform you that your manuscript has been deemed suitable for publication in PLOS ONE. Congratulations! Your manuscript is now with our production department. 

Kind regards, 

on behalf of

Prof. Christof Markus Aegerter 

Section Editor

PLOS ONE